# Miniaturised interaction proteomics on a microfluidic platform with ultra-low input requirements

Cristina Furlan [1], René A.M. Dirks[1], Peter C. Thomas[2], Robert C. Jones [2], Jing Wang[2], Mark Lynch[2], Hendrik Marks [1] & Michiel Vermeulen [3]

Essentially all cellular processes are orchestrated by protein-protein interactions (PPIs). In recent years, affinity purification coupled to mass spectrometry (AP-MS) has been the preferred method to identify cellular PPIs. Here we present a microfluidic-based AP-MS workflow, called on-chip AP-MS, to identify PPIs using minute amounts of input material. By using this automated platform we purify the human Cohesin, CCC and Mediator complexes from as little as 4 micrograms of input lysate, representing a 50—100-fold downscaling compared to regular microcentrifuge tube-based protocols. We show that our platform can be used to affinity purify tagged baits as well as native cellular proteins and their interaction partners. As such, our method holds great promise for future biological and clinical AP-MS applications in which sample amounts are limited.

[1] Department of Molecular Biology, Faculty of Science, Radboud Institute for Molecular Life Sciences, Radboud University Nijmegen, Nijmegen 6525 GA, The Netherlands. [2] Fluidigm Corporation, South San Francisco, CA 94080, USA. [3] Department of Molecular Biology, Faculty of Science, Radboud Institute for Molecular Life Sciences, Oncode Institute, Radboud University Nijmegen, Nijmegen 6525 GA, The Netherlands. These authors contributed equally: Cristina Furlan, René A.M. Dirks. These authors jointly supervised this work: Hendrik Marks, Michiel Vermeulen. Correspondence and requests for materials should be addressed to H.M. (email: H.Marks@ncmls.ru.nl) or to M.V. (email: Michiel.Vermeulen@science.ru.nl)

dentification and characterisation of protein–protein interactions (PPIs) is a major focus area in biology. A variety of approaches have been developed to study PPIs, such as yeast two-hybrid screening, co-immunoprecipitation combined with western blots and phage display[1,2].

During the last 2 decades, affinity purification mass spectrometry (AP-MS) has proven to be a powerful technology to elucidate protein–protein interactions in cells and tissues[3–5]. Antibodies targeting native proteins can be used to investigate protein interactions on a large scale[6]. When highly specific antibodies are not available, epitope tagging is an attractive alternative for AP-MS. In addition, complementary approaches to identify PPIs have been developed, such as proximity labeling technologies[7–9]. In all these studies, up to milligrams of lysates isolated from millions of cells are used for each affinity purification. For this reason, AP-MS screenings are typically performed on immortalized cells such as cancer cell lines which can easily be grown in very large quantities and which can be genetically manipulated[10,11].

In recent years it has become apparent that PPIs and their dynamics are important during development[12]. Furthermore, perturbed PPIs play critical roles in establishing disease phenotypes (e.g., in amyloidosis)[13–16]. Therefore, it is increasingly important to identify PPIs in near-physiological models (i.e., in organoids or early embryonic tissue) and in clinical samples. Those specimens are often available in limited quantities and are therefore not applicable for interaction proteomics studies.

Here, we describe a microfluidic-based AP-MS platform called on-chip AP-MS which facilitates PPI studies from as little as 12,000 input cells, representing a 50–100-fold downscaling compared to regular microcentrifuge tube-based workflows. We benchmark our method using a number of well-characterized mammalian protein complexes, which we purified from green fluorescent protein (GFP)-tagged HeLa cells. In addition, we provide evidence that our platform can be used to purify cellular protein complexes using antibodies against native proteins.

## Results

**Developing and benchmarking on-chip AP-MS.** To explore the input limitations of microcentrifuge-based AP-MS workflows, we performed preliminary affinity purification experiments using transgenic HeLa Kyoto cell lines expressing SMC1A as a GFP fusion protein at near-endogenous levels using BAC TransgeneOmics[17]. SMC1A is a subunit of the human Cohesin complex, which plays an important role in chromosome segregation, DNA repair, and transcriptional regulation. In the presence of ATP, SMC1A forms a heterodimer with SMC3. The SMC1A/SMC3 dimer interacts with RAD21 and STAG1/2 to form a stable Cohesin ring around DNA[18]. We purified SMC1A-GFP from a range of input lysates starting from 500 μg of whole cell extract down to 4 μg. As a negative control, we used wild-type HeLa Kyoto cells. These experiments revealed that decreasing the amount of input lysate in microcentrifuge tube-based affinity purification workflows results in a gradual loss of performance and detection of interaction partners. At the lowest titration point, 4 μg of input whole-cell extract, detection of almost all interaction partners is lost (Supplementary Figure 1, Supplementary Data 1). These results illustrate the urgent need to develop alternative AP-MS protocols that are compatible with low-input sample amounts. Previous studies have made initial steps in this direction, for example, by establishing AP-MS protocols on 96-well filter plates, but those approaches still require several hundreds of micrograms of input lysate[19–21].

We set out to develop an AP-MS workflow that enables robust detection of PPIs from small amounts of whole-cell lysate. To

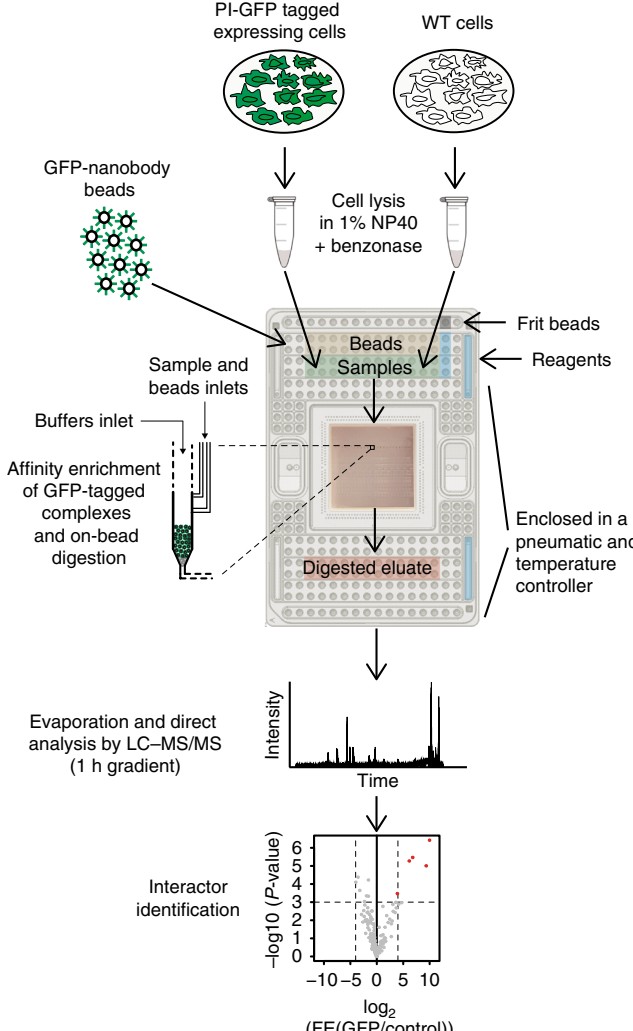

**Fig. 1** Workflow for affinity purification of GFP-tagged complexes on a microfluidic device. A limited number of cells (in this paper we used 12,000 cells for proof-of-principle) expressing a GFP-tagged form of a protein of interest (PI) are lysed to obtain whole cell extracts that serve as pools of proteins. Test samples as well as the appropriate controls are loaded into a microfluidic chip which is preloaded with all buffers and reagents that are required during the IP and preparation of the samples for the LC–MS/MS. During the microfluidic run, columns of GFP-nanobody beads are packed in single reactors and subsequently washed before loading of the protein samples from the inlet onto the columns. Tagged macromolecular assemblies are retained on the GFP-nanobody bead columns and digested to peptides. Peptide eluates are recovered and prepared for mass spectrometry analysis, after which specific interactors are identified

achieve this, we took advantage of an analog microfluidic system containing purification columns to perform affinity purifications followed by on-bead protein digestion (Fig. 1). This all occurs in very small volumes (nanoliter scale) and in a fully automated manner, thereby reducing variation between experiments. Notably, the development of the protocol involved assessing compatibility of reagents[22] and affinity purification steps with the microfluidic platform. To allow loading of the samples in the microfluidics channels and/or subsequent washing of the columns on the microfluidic platform, we incorporate two important adaptations as compared to regular pull-down protocols: exclusion of glycerol as cryoprotectant in the extract preparation and removal of ethidium bromide as competitor for

protein–DNA interactions in extract incubation with the beads. We call this method on-chip AP-MS.

To benchmark the on-chip AP-MS method, we made use of the before-mentioned SMC1A-GFP cell line and wild-type HeLa Kyoto cells as control. Whole-cell extracts from these cell lines (4 µg per pull-down in triplicate) were loaded by pressure-driven laminar flow on a prototypical microfluidic chip containing 24 parallel columns packed with beads covered with epoxy groups functionalized with an in-house purified GFP nanobody[23]. After loading lysates, columns were washed to remove background proteins, after which column-bound proteins were on-bead digested with trypsin. The resulting peptide mixtures were then recovered from the microfluidic chip and subjected to liquid chromatography–tandem mass spectrometry (LC–MS/MS) analysis on a high-performance hybrid Orbitrap mass spectrometer. A label-free quantification method with $t$ test-based statistics was used for interactor identification. As shown in Fig. 2a, loading 4 µg of SMC1A-GFP extract obtained from bulk preparation of cells on the microfluidics chip yielded all known core Cohesin subunits as statistically significant interaction partners (Fig. 2a, Supplementary Data 2). To probe for sensitivity, we prepared whole-cell extracts from different amounts of input cells and then loaded 4 µg of extract from these preparations on the microfluidics chip. We did not detect a loss in sensitivity when decreasing the number of input cells (Supplementary Figure 2, Supplementary Data 3). We were even able to detect all core SMC1A interactors starting from ~$12 \times 10^3$ input cells, roughly equivalent to 4 µg of whole-cell extract, per affinity purification (Fig. 2b; individual replicates in Supplementary Figure 3, Supplementary Data 4). Importantly, the purified complex from these low-input samples showed a similar stoichiometry compared to regular affinity purifications performed with 500 µg of input lysate (Fig. 2c; individual replicates in Supplementary figure 3). However, it should be noted that even though we identified well-characterized Cohesin complex interactors (WAPL, PDS5A/PDS5B) as substoichiometric SMC1A-GFP interactors when using 500 µg of input lysate (Supplementary Figure 1), these proteins are not identified in low input SMC1A-GFP pull-downs. Based on these results we conclude that our low-input AP-MS method has its limitations regarding the ability to detect low-abundant substoichiometric (<0.1 relative to the bait) and/or dynamic interactions between core subunits and regulatory proteins for any given protein complex, depending on the stoichiometry, affinity, and copy number of the bait and its interaction partners.

**On-chip AP-MS on low-abundant complexes**. Cohesin is an abundant nuclear complex, with the four most prominent subunits expressed between $2.8 \times 10^5$ and $4 \times 10^5$ protein copies per cell[10]. To further investigate the sensitivity of our microfluidic platform, we tested whether a lower abundant complex containing more subunits could be successfully identified. The coiled-coil domain-containing protein 93 (CCDC93) is present at ~$65 \times 10^3$ copies per cell[10]. CCDC93 is part of the CCC complex composed of COMMD proteins, CCDC22 and CCDC93[24]. The CCC complex, residing in the cytosol and recruited to endosomes by FAM21, regulates copper trafficking[25]. By performing on-chip AP-MS on CCDC93-GFP expressing cells, we identified all known CCC complex members as significant interaction partners, both by using 4 µg of whole-cell extract from a large bulk extraction and from $12 \times 10^3$ input cells per pull-down (Fig. 2d, e; individual replicates in Supplementary Figure 3, Supplementary Datas 5 and 6). The stoichiometry of these interactions are largely in line with values obtained using conventional high-input workflows[10] (Fig. 2f).

To further benchmark our on-chip AP-MS workflow we purified the Mediator complex, a very large, well-characterized complex (~1.4 MDa) involved in gene regulation[26,27]. Mediator complex subunits are expressed at cellular copy numbers ranging from ~850 to $95 \times 10^3$ in HeLa cells[10]. We purified the Mediator complex from CDK8-GFP (copy number ~$38 \times 10^3$ per cell[10]) expressing HeLa Kyoto cells using $25 \times 10^3$ input cells per pull-down. We retrieved essentially all (>25) human Mediator subunits as statistically significant CDK8 interactors, albeit at lower stoichiometry as compared to high-input workflows[10] (Fig. 3a, b, Supplementary Data 7). Based on these results we conclude that, whereas our on-chip-AP-MS workflow is able to confidently identify PPIs for relatively low-abundant baits and interaction partners, stoichiometry estimations for these interactions are more challenging. Nevertheless, these three examples illustrate that we established a microfluidic-based AP-MS platform that requires ~50–100-fold less input extract than conventional workflows and is thus compatible with clinical samples or rare cell types that are only available in very small quantities. To illustrate this, we purified the GFP-fused mitotic spindle checkpoint protein BUBR1-GFP from $12 \times 10^3$ G2/M sorted cells, and identified its dimerization partner, BUB3 (Supplementary Figure 4, Supplementary Data 8). Using a conventional purification strategy and in the absence of any drugs to arrest cells in mitosis, this would have required at least $10^7$ input cells per technical replicate.

**Expanding the applicability of on-chip AP-MS**. For increased throughput, reduced loading times and the capability of using a larger array of buffers to facilitate more complex AP-MS workflows, we performed the GFP-SMC1A and GFP-CCDC93 affinity purifications on a redesigned microfluidic chip with 48 parallel reactors and obtained similar results (Supplementary Figure 5, Supplementary Data 9). We further used this platform to purify protein complexes using antibodies against endogenous proteins in conjunction with protein A/G beads. We were able to purify SMC1A-GFP using a polyclonal GFP antibody rather than a GFP nanobody, indicating that our workflow can be used to purify endogenous protein complexes as well (Fig. 4a). To further support these observations, we also purified the human Cohesin complex from HeLa Kyoto cells using an antibody targeting native endogenous SMC3 (Fig. 4b, Supplementary Data 10). The performance of these endogenous protein pull-downs will obviously greatly depend on the antibody affinity and specificity, and the abundance of the protein complex. The current workflow can be conveniently applied on the commercial Fluidigm C1 platform, as exemplified in this study. We anticipate that the protocol presented here can also be easily adapted to other programmable microfluidic platforms with a similar design, namely nanoliter-sized affinity purification columns targeting tagged, proximity labeled or native proteins with pressure-driven laminar flow of buffers and lysates. However, it should be noted that the design of the chips that we employed for our experiments requires using beads with a size between 2 and 10 µm. This currently excludes the use of some commonly utilized affinity resins, such as Streptactin[28], and Flag/HA, as these are not available in this size range at this point in time[29].

**Discussion**

With on-chip AP-MS we made important and much-needed steps to facilitate AP-MS for low-input samples, namely microfluidics-based miniaturization and automation. However, further improvements to the workflow, including direct cell lysis in the microfluidic chip and coupling of the chip to the LC of the mass spectrometer might further reduce input requirements and

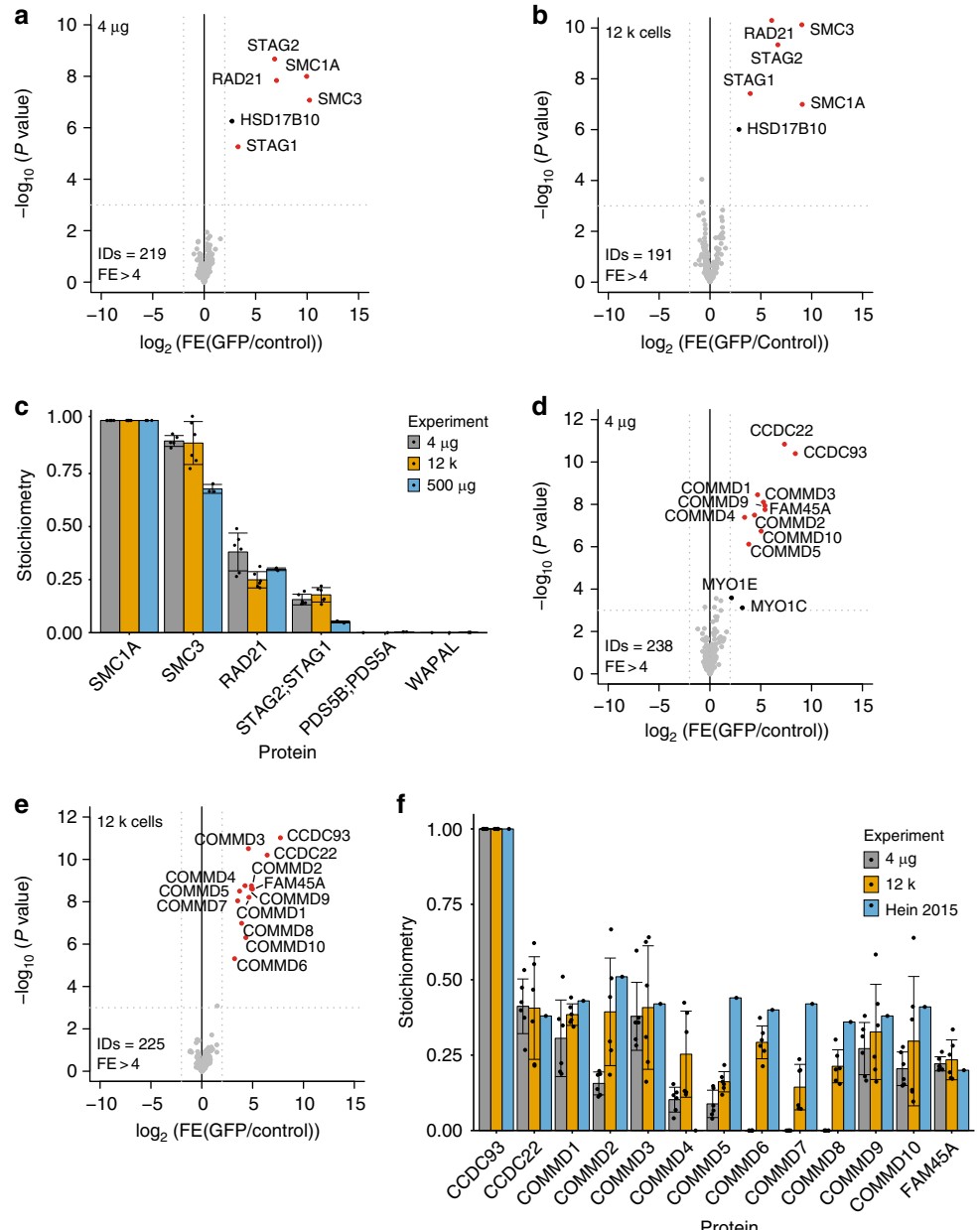

**Fig. 2** Benchmark for on-chip AP-MS on low amounts of total extract using SMC1A-GFP and CCDC93-GFP baits. **a**, **b** Volcano plots from label-free microfluidic pull-downs for SMC1A-GFP bait using 4 μg (**a**) or 12,000 cells (**b**). The enrichment of the bait and its partners as fold enrichment LFQ GFP over LFQ control (x-axis) is plotted against the $-\log_{10}$ transformed $P$ value of the permutation based FDR corrected $t$ test (y-axis). ID refers to the total number of proteins reported and FE stands for fold enrichment. Dotted gray lines represent statistical cut-offs. Black dots identify significant proteins; red dots are significant proteins that are known interactors of the bait. HSD17B10 is a known contaminant of pull-downs using Dynabeads®[33]. Each volcano plot presents $n = 2$ biologically independent experiments, each comprised of $n = 3$ technical replicates for PI-GFP and control extract. **c** Stoichiometry of the Cohesin subunits as compared to SMC1A for the experiments shown in panels a and b, as well as a 500 μg microcentrifuge tubes-based pull-down shown as gold standard (data from Supplementary figure 1a). iBAQ values for each of the interaction partners of Cohesin were divided by the iBAQ values of the bait, so that is set to 1. Data display mean values ± s.d. Gray bars represent the stoichiometry of Cohesin complex as derived from for the 4 μg input experiment, yellow bars for the 12,000 cells input experiment and light blue bars for the 500 μg input microcentrifuge tubes-based pull-down. Black dots represent the individual data points. **d**, **e** Volcano plot from microfluidic pull-down for CCDC93-GFP bait using 4 μg (**d**) and 12,000 cells (**e**). Dotted gray lines represent statistical cut-offs. Black dots identify significant proteins; red dots are significant proteins that are known interactors of the bait. MYO1E and MYO1C are known contaminants[33]. ID refers to the total number of proteins reported and FE stands for fold enrichment. Each volcano plot presents $n = 2$ biological replicates, each comprised of $n = 3$ technical replicates for PI-GFP and control extract. **f** Stoichiometry of the CCC complex subunits as compared to CCDC93 for the experiments shown in panels (**d**) and (**e**), using the pull-down performed by Hein et al.[10] as reference. iBAQ values for each of the CCC interaction partners were divided by the iBAQ values of the bait (set to 1). Data display mean values ± s.d. Gray bars represent the stoichiometry of CCC complex as derived from the 4 μg experiment, yellow bars for the 12,000 cells experiment and light blue bars for microcentrifuge tube-based pull-down performed by Hein et al.[10]. Black dots represent the individual data points. Source data are provided as a Source Data file

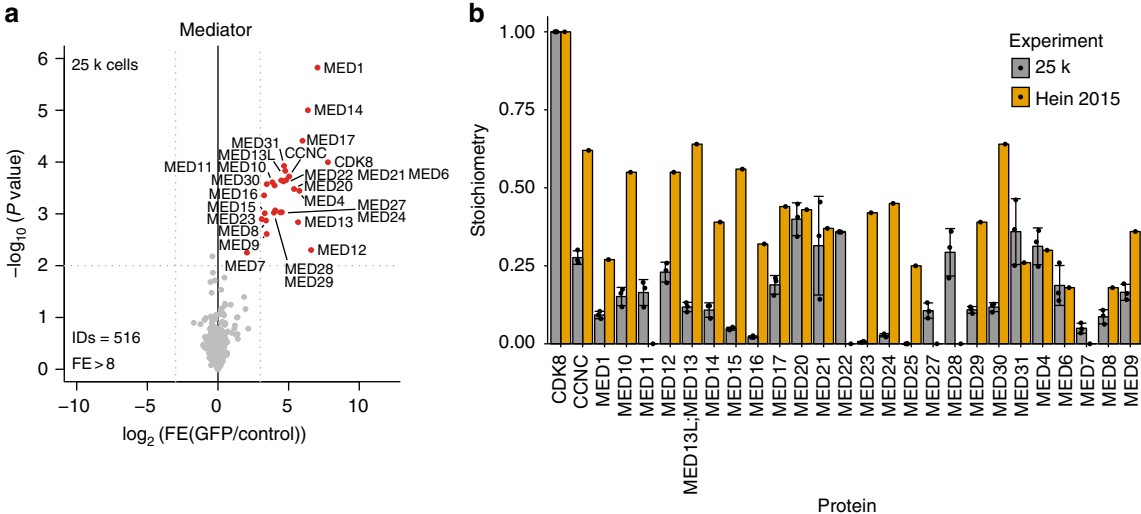

**Fig. 3** Benchmark of on-chip AP-MS with Mediator complex. **a** Volcano plot from label-free microfluidic pull-down for CDK8-GFP bait using 25,000 cells. Dotted gray lines represent statistical cut-offs. Black dots identify significant proteins; red dots are used for known interactors of the bait. ID refers to the total number of proteins reported and FE stands for fold enrichment. The volcano plot presents $n = 1$ biologically independent experiment, each comprised of $n = 3$ technical replicates for PI-GFP and control extract. **b** Stoichiometry of the mediator subunits as compared to CDK8 for the experiments using the same pull-down performed by Hein et al.[10] as reference. iBAQ values for each of the interaction partners of Mediator complex were divided by the iBAQ values of the bait (CDK8, set to 1). Data display mean values ± s.d. Gray filled bars are used for representing the Mediator complex stoichiometry as obtained by the 25,000 cells input experiment while yellow bars describe the stoichiometry for Mediator obtained by the microcentrifuge tube-based pull-down performed by Hein et al.[10]. Black dots represent the individual data points. Source data are provided as a Source Data file

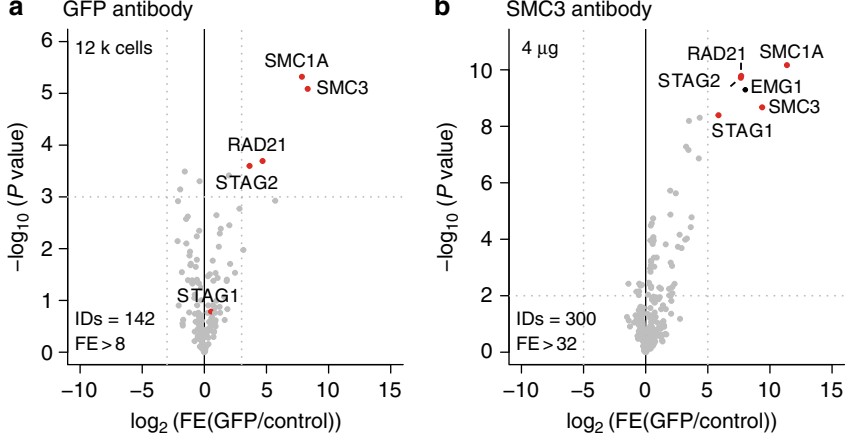

**Fig. 4** On-chip AP-MS using antibodies. **a**, **b** Volcano plot of SMC1A-GFP enriched using a GFP antibody (**a**) and an antibody against native SMC3 protein (**b**). ID refers to the total number of proteins reported and FE stands for fold enrichment. Dotted gray lines represent statistical cut-offs. Black dots identify significant proteins; red dots are used for known interactors of the bait. EMG1 is a known contaminant[33]. The volcano plot in panel a presents $n = 1$ biologically independent experiment, each comprised of $n = 3$ technical replicates for PI-GFP and control extract. The volcano plot in panel (**b**) presents $n = 2$ biological replicates, each comprised of $n = 3$ technical replicates for PI and control extract

hands-on time, while increasing robustness and sensitivity of our on-chip AP-MS workflow. Altogether, we envision that the technology that we present here and improvements thereof could, in principle, be applied to even lower cell numbers or, eventually, single cells. In any case, with the development of on-chip AP-MS, the field of mass spectrometry-based interaction proteomics is taking the next step towards applications in biology and medicine, which thus far, due to technical constraints, have remained elusive.

## Methods
**Cell culture and whole cell extract preparation**. For preparing the extract for on-chip AP-MS, HeLa Kyoto WT and BAC lines expressing a C-terminal LAP tagged form of SMC1A, CCDC93 and CDK8 were obtained from the Hyman laboratory (Max Planck Institute of Molecular Cell Biology and Genetics, Dresden, Germany) by courtesy of Dr. Ina Poser. Cells were cultured in high-glucose Dulbecco's modified Eagle medium (DMEM) supplemented with 10% fetal bovine serum (FBS) and 100 U/mL penicillin and streptomycin. BAC lines were kept under selection by addition of 400 μg/μL geneticin (G418). All reagents for cell culture were purchased from Gibco, Life Technologies™. Cells were harvested by trypsinization and centrifuged at 400×g for 5 min at 4 °C. Cell pellets were washed once in PBS. An aliquot of cells was stained with Trypan Blue 0.4% (Sigma Aldrich, Merck) and counted on an automated cell counter (TC10™, BioRad or Countess II, Invitrogen, Thermo Fisher Scientific). The number of cells required was pipetted into a 0.5 or 1.5 mL Protein LoBind tube (Eppendorf) and centrifuged again for 5 min at 400×g at 4 °C. PBS was removed using a 30 g needle connected to a vacuum aspiration system. Cells were lysed in a buffer containing 150 mM NaCl, 50 mM Tris-HCl pH 8, 1 mM MgCl₂, 250 U/mL Benzonase® (Novagen, Millipore), EDTA-free complete proteins inhibitors (Roche®), and 1% NP-40, 1 mM DTT.

Lysis was performed using (i) 1 μL of buffer per 25,000 cells; or (ii) 1 μL of lysis buffer for extracting cell numbers lower than 25,000 cells. Samples were incubated at 4 °C for 2 h rotating (or rocking for low-volume samples) on a Intelli-Mixer RM-2L program uu60 (Elmi). The extracts were spun at 4000×g for 30 min at 4 °C. When extracting a large batch of cells, the supernatant was collected without the lipid layer. Cleared lysates were transferred to another tube and adjusted to a final volume of 2.5 μL with lysis buffer. To further eliminate aggregates, lysates were spun down 2 min at 18,800 × g at 4 °C on a table centrifuge right before they were loaded onto the microfluidic device. Protein concentration for bulk lysates obtained from a large number of cells was assessed using a BCA assay (Thermo Fisher Scientific). The equivalent of 4 μg of proteins was loaded using a 2.5 μL volume.

For preparing the extract for microcentrifuge tube-based pull-down, cells were washed in PBS and harvested as previously described. Pellets were resuspended in 5 cell pellet volumes of lysis buffer containing 150 mM NaCl, 50 mM Tris-HCl pH 8, 1 mM EDTA, 20% glycerol, EDTA-free complete proteins inhibitors (Roche), 1% NP-40, and 1 mM DTT. Lysates were incubated at 4 °C for 2 h on a rotating wheel and subsequently centrifuged for 30 min at 4000×g at 4 °C. The supernatant was collected and snap-frozen in liquid nitrogen.

**Cell culture and sorting.** HeLa-FlpIn TRex cells expressing LAP-BUBR1 obtained from the Kops laboratory (Hubrecht Institute, Utrecht, The Netherlands) were grown in the same conditions as the HeLa BAC lines but puromycin (1 μg/mL) was used to maintain selection. Cells were induced for ~48 h with doxycyclin (1 μg/mL). For cell cycle sorting based on DNA content, 8–9 × 10^6 of induced and not induced cells were harvested and stained in medium with 5 μM Hoechst 33342 (Thermo Scientific) for 60 min at 37 °C. Cells were centrifuged at 400×g and resuspended in PBS containing 1% FBS at a concentration 30 × 10^6 cells/mL. Shortly before sorting, cells were strained on a 70 μm sterile Syringe Filcons (BD biosciences). Sorting was performed on a BD FACSAria instrument and 12 × 10^3 cells were collected in tubes containing 200 μL of PBS–1% FBS. Lysates were prepared as above-mentioned.

**Purification of anti-GFP nanobodies.** Purification of the lysine free anti-GFP nanobody (Lag16-2K/R) was performed as previously described[17]. Briefly, plasmid containing the Lag16-2K/R nanobody sequence was transformed into Arctic Express (DE3) competent cells. Transformed bacteria were grown overnight in LB containing 100 μg/ml ampicillin and 20 μg/ml gentamicin (Sigma G1272). Totally, 10 ml of the overnight culture was used to expand the culture to OD_600 0.6 in 1 l LB containing 100 μg/ml ampicillin. Nanobody expression was induced by 0.1 mM Isopropil-β-D-1-tiogalattopiranoside at 12 °C for ~24 h. Cells were then harvested by centrifugation and periplasmic extract was prepared incubating the pellet with TES buffer (0.2 M Tris-HCl pH 8, 0.5 mM EDTA, 0.5 M sucrose) for 30 min on ice. Extract was cleared by sequential centrifugations firstly for 15 min at 4500×g and secondly for 30 min at 38,800×g. Final supernatant was conditioned to 0.15 M NaCl and incubated for 30 min at 4 °C with 2 ml of HIS-Select® Nickel Affinity Gel (Sigma-Aldrich®, P6611) previously washed with water and equilibrated with wash buffer I (20 mM sodium phosphate buffer pH 8, 0.9 M NaCl). After incubation the resin was initially washed with 6 volumes of wash buffer I, then with 6 volumes of wash buffer II (20 mM sodium phosphate buffer pH 8, 0.15 M NaCl, 10 mM imidazole pH 8) and finally His-purified nanobody was eluted with 4 column volumes of elution buffer (20 mM sodium phosphate buffer pH 8, 0.15 M NaCl, 250 mM imidazole pH 8). Eluted nanobody was dialyzed overnight in PBS in a Slide-A-Lyzer™ G2 Dialysis Cassettes, 10 K MWCO, 5–15 mL (Thermo Fisher Scientific, 87,731). Finally, nanobody was concentrated using a Centriprep Centrifugal Filter Unit 3 kDa cutoff, 15 mL volume (Millipore, 4303).

**Chemical coupling of anti-GFP nanobodies to magnetic beads.** Chemical conjugation of the anti-GFP nanobody to epoxyl Dynabeads® M-270 beads (Invitrogen, Thermo Fisher Scientific) was performed as previously described[22]. Briefly, 10 mg of epoxyl Dynabeads® M-270 beads were conjugated with 100 μg Lag16-2K/R anti-GFP nanobody. Beads were washed for 15 min with 0.1 M sodium phosphate buffer pH 7.4 and for 5 min with 0.1 M sodium phosphate buffer pH 8 rocking. Antibody mix was prepared by adding 100 μg nanobody, 0.1 M sodium phosphate buffer pH 8–200 μl final volume and 1 M ammonium sulfate in 0.1 M sodium sulfate pH 7.4 (added dropwise). Antibody mix was centrifuged at maximum speed on a bench top centrifuge for 5 min before being added to the washed beads. Antibody mix and beads were incubated ~20 h at 30 °C tilted-rotating in a round bottom 2 ml microcentrifuge tube. After incubation conjugated beads were washed with 100 mM glycine pH 2.5, 10 mM Tris-HCl pH 8.8 and 100 mM triethylamine freshly prepared to quench the reaction, wash away unbound nanobody and block unreacted epoxyl groups on the beads. Finally, anti-GFP conjugated beads were washed with PBS for 5 min 4 times, with 0.5% NP40 in PBS one time for 5 min and a second time for 15 min rocking, and with PBS for 5 min before storage at −20 °C in 30% glycerol and 0.02% sodium azide in PBS until further use.

**Antibody-based GFP or endogenous protein pull-down.** Extracts from 12 × 10^3 cells were prepared as previously described. Two microlitre of extract were incubated with 180 ng of GFP antibody (HPLC purified Abcam Ab290: Ab6556) or Rabbit Control IgG (Abcam 46540, Lot.GR63822-2, ChIP grade). Extracts from

HeLa Kyoto cells were incubated with anti-SMC3 antibody (Abcam Ab9263, ChIP grade) or normal rabbit IgG (Santa Cruz sc-2027) at 90 ng/4 μg antibody/extract ratio. Extract and antibody were incubated for 1 h at 4 °C before proceeding with loading into the microfluidics device.

**GFP pull-down protocol on microfluidic device.** Two types of poly-dimethylsiloxane microfluidic chips were manufactured and provided by Fluidigm® (Fluidigm Corporation, South San Francisco, CA, USA). One is a prototypical 24 reactor chip and the other is a final 48 reactor chip. The chips are mounted to a carrier that facilitates loading of testing and control samples, anti-GFP nanobody beads, frit beads and the buffers required to perform the on-bead pull-down. Pressure control and thermocycling were automated using the C1™ system (Fluidigm®). All reagent and valve control channels were dead-end filled before script operation to avoid the presence of air bubbles in the system during the workflow. The procedure is started by loading a mixture of 4.5 and 6.0 μm frit beads packed into the reactor at 13 psi for 45 s to reduce the size of the reactor exit (5 μm drain opening size) to enable building of the column. For the 24-parallel reactor chip, anti-GFP nanobody 2.8 μm beads were packed for 10 min at 13 psi in a 23 nL reactor to achieve a column size of 5 nL; beads concentration was 15 × 10^7 beads/mL in 30% glycerol. On the redesigned 48 parallel reactor final chip, beads are packed in 6 cycles to obtain a column of equal capacity. Bead columns on both designs were washed with high salt buffer [1 M NaCl, 20 mM Hepes pH 7.9, 2 mM MgCl_2, 0.2 mM EDTA, EDTA-free complete proteins inhibitors (Roche®), 1% NP-40, 0.5 mM DTT]. Washing was performed by building up 11 psi of pressure across the length of the reactors, then closing the buffer entry channel followed by the release of the pressure toward a waste container. The columns were washed for 15 cycles before cell extracts were loaded. Whole-cell extract samples were loaded across the bead columns for 20–60 min using 5–12 psi pressure at 15 °C. All subsequent washes were also performed using 15 cycles to build up a pressure of 11 psi over the reactor; which guaranteed the transit of approximately three reactor volumes of buffer to pass through the column. After incubation of the extracts, the reactors were washed with high salt buffer followed by a wash with PBS, both at 10 °C. To proceed with on-beads protein digestion, the temperature was increased to 37 °C. PBS was displaced by 10 mM DTT in digestion buffer [25 mM ammonium bicarbonate, 10% acetonitrile], after which the bead columns were incubated with reduction buffer for 30 min. The chip temperature was increased to 22 °C and reduction buffer was replaced by alkylation buffer [55 mM iodoacetamide in digestion buffer], after which the bead columns were incubated with alkylation buffer for 20 min. After bringing the chip temperature to 37 °C, trypsin buffer [0.005 μg/μL trypsin (Promega™) in digestion buffer] was introduced and kept in the reactors for 90 min. Peptides were harvested at 25 °C using 300 cycles of 13 psi built-up pressure over the length of the chip (approximately, 100 nL volume), yielding a total of 3–3.5 μL of eluate for each reaction. All valves were locked before disconnecting and releasing the chip from the C1 device. Eluted peptides were recovered in PCR tubes compatible with loading into the LC auto sampler for mass spectrometry.

**GFP pull-down protocol in tubes.** GFP affinity enrichment was performed using GFP-Trap®_A beads (ChromoTek). Briefly, 15 μL of beads from a 50% slurry were washed three times with incubation buffer [300 mM NaCl, 20 mM HEPES–KOH (pH 7.9), 20% glycerol, 2 mM MgCl_2, 0.2 mM EDTA, EDTA-free complete proteins inhibitors (Roche®), 1% NP-40, 0.5 mM DTT 1% NP-40, 0.5 mM DDT]. All washes were performed by adding 1 mL of buffer, rotating the sample 10 times to resuspend the beads and spinning for 2 min at 1500×g at 4 °C before removal of the buffer. Beads (7.5 μL) were combined with 500, 100, 20, and 4 μg of whole-cell extract from SMC1A BAC and HeLaK control cell lines. Incubation was performed in 400 μL final volume of incubation buffer containing ethidium bromide at a final concentration of 50 μg/μL for 90 min at 4 °C. Beads were then washed twice with a high-salt buffer [1 M NaCl, 20 mM Hepes pH 7.9, 2 mM MgCl_2, 0.2 mM EDTA, EDTA-free complete proteins inhibitors (Roche®), 1% NP-40, 0.5 mM DTT], followed by 2 washes with PBS containing 1% NP-40 and finally 3 washes with PBS. On-bead digestion of proteins and the following desalting and purification of peptides for mass spectrometry analysis were performed as in Baymaz et al.[30]. Briefly liquid-free beads were resuspended in 50 μl elution buffer (2 M urea, 100 mM Tris HCl pH8, 10 mM DTT) and incubated at RT for 20 min shaking. After that 50 mM iodoacetammide was added to the bead suspension and incubated for 10 min shaking in the dark. 0.25 μg trypsin (Promega™) were added to the beads suspension and incubated for 2 h at 25 °C shaking vigorously. After incubation time bead suspension was centrifuged, supernatant was moved to another microcentrifuge tube and beads were washes again with 50 μl elution buffer and incubated for 5 min shaking. Supernatant containing the digested proteins were combined and 0.1 μg trypsin was added before over night incubation at RT. Peptide solution was then cleaned over a C18 column using the stage-tip protocol. In short columns were conditioned with 5 μl methanol and washed with 20–50 μl 0.1% formic acid in water (FA). Trypsin digestion was blocked by adding 50 μl of 0.1% FA and 5 μl 5% trifluoroacetic acid to reduce pH conditions. Peptides were then bound to the C18 column, washed with 20–50 μl 0.1% FA and stored at 4 or −20 °C until mass spectrometry analysis. At that point peptides were eluted with 30 μl 80% acetonitrile in 0.1% FA and evaporated on a Concentrator Plus (Eppendorf) using the program V-AQ for 15 min and were reconstituted to a 12 μL volume with 0.1% FA.

**MS and data analyses**. Digestion buffer (10 or 25 µL) was added to the on-chip AP-MS eluted peptides and the solution was incubated overnight at 37 °C in a wet chamber. Overnight digestion solutions were acidified by adding 1 µL of 10% formic acid first and diluted with an additional 25 µL of 0.1% formic acid. Samples were evaporated on a Concentrator Plus (Eppendorf) using the program V-AQ for 60–100 min and were reconstituted to a 7–8 µL volume with 0.1% formic acid. Samples were either directly measured with a single injection of 5 µL on the mass spectrometer or stored at −80 °C until analysis by LC–MS/MS. Chromatography separations and mass spectrometry analyses were performed on an Easy-nLC™ 1000 coupled to a LTQ-Orbitrap Fusion™ Tribrid™ (Thermo Fisher Scientific). Peptides were separated using an organic gradient obtained by mixing Buffer A (0.1% formic acid in LC grade water) and Buffer B (80% acetonitrile, 0.1% formic acid in LC grade water). The linear gradient was formed over 33 min starting from 15% Buffer B and ending to 35% Buffer B. The reverse phase column was then washed with 60 and 90% Buffer B for 5 min each buffer and re-equilibrated to 5% Buffer B for a total acquisition time of 65 min. The flow rate was 200 nL/min. Analytical columns for reverse phase separations were packed in house with C18 material [ReproSil-Pur C18-AQ particle size 1.9 µm (Dr. Maisch GmbH, Germany)] in a 300 mm long column spray emitter by means of an air-pressure pump (Next Advance, Inc.). Emitters were 360 µm OD, 75 µm ID with an opening of $8 \pm 1$ µm (New Objectives, Inc.). Data were acquired in data-dependent top speed mode excluding +1 and peptides with unassigned charge and including charges up to 7+. Peptide full spectra were recorded from 300 to 1600 m/z on the Orbitrap mass analyzer set at 120 k resolution in profile mode, using an AGC target of 2E5, a maximum injection time of 50 milliseconds, and an exclusion time of 1 min. MS/MS spectra were acquired in the linear ion trap mass analyzer using CID for fragmentation with a collision energy at 35%, a stepped collision energy of 5; resolution was set to 30 k, AGC target to 1.5E4, scan rate to rapid, and intensity threshold to 5E4. Data searches were run against SwissProt human database (comprising reviewed entries only and downloaded in June 2017) using standard settings on MaxQuant software (version 1.5.1.0). Briefly, 1% false-discovery rate (FDR) was applied to the match of propensity-score matching and assembly of proteins. Mass tolerance for correct matches was set to 20 ppm for Fourier transform mass spectrometry analyzer and 0.5 Da for ion trap mass spectrometry MS/MS matches. Carbamidomethylation of cysteines was included as fixed modification while acetyl at protein terminus and methionine oxidation were considered variable modifications. Two missed cleavages were allowed for trypsin enzyme cuts and peptides length was set between 1 and 7 aminoacids. We performed label free quantification with option match between runs and iBAQ quantification of proteins. Perseus software (version 1.5.5.3) was used to perform filtering, imputation of missing values from a normal distribution, and permutation-based FDR corrected $t$ test. The data obtained were visualized by R. Relative stoichiometry of complexes was calculated as in Smits et al.[31]. The MS proteomics data have been deposited to the ProteomeXchange Consortium via the PRIDE[32] partner repository with the dataset identifier PXD012800.

**Reporting Summary**. Further information on experimental design is available in the Nature Research Reporting Summary linked to this article.

## Data availability

The data sets generated during and/or analyzed during the current study are available in the PRIDE repository with the accession number PXD012800 [http://proteomecentral.proteomexchange.org/cgi/GetDataset?ID = PXD012800].
The source data underlying Figs. 2c, f and 3b and Supplementary Figs. 1e and 2d are provided as a Source Data file. A reporting summary for this Article is available as a Supplementary Information file. All other data supporting the findings of this study are available from the corresponding authors upon reasonable request.

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

## Acknowledgements

We would like to thank all previous and current members of the Vermeulen and Marks groups for fruitful discussions and feedback, Prof. Geert Kops for kindly donating the LAP-BUBR1 HeLa cells and Dr Ina Poser for providing us with the BAC cell lines. We

also thank Rob Woestenenk at the RadboudUMC flow cytometry facility for assistance with the LAP-BUBR1 cells sorting and Marijke Baltissen for technical support during revision experiments. Funding for M.V. was provided by the Netherlands Organization for Scientific Research (NWO Gravitation Program Cancer Genomics Netherlands) and by an ERC Consolidator Grant (771059). The Vermeulen lab is part of the Oncode Institute, which is partly financed by the Dutch Cancer Society (KWF). H.M. is supported by the Netherlands Organization for Scientific Research (NWO-VIDI 864.12.007).

## Author contributions

C.F. and M.V. conceived the study. H.M. and R.A.D. provided expertise in the microfluidic technologies, C.F. and M.V. provided expertise in the AP-MS. C.F. and R.A.D. performed the experiments, C.F analyzed the MS data, R.A.D. repurposed and programmed the microfluidic platform, P.T., R.C.G., J.W., and M.L. provided support with the microfluidic platform. C.F. and M.V. wrote the paper with input from R.A.D. and H.M. H.M. coordinated the project. H.M. and M.V. designed the study and supervised the project.

## Additional information

**Competing interests:** J.W. and M.L. are current employees of Fluidigm Corporation and may hold stock in the company. P.T. and R.J. are former employees of Fluidigm Corporation and may still hold stock in the company. The remaining authors declare no competing interests.

