## [Peer Review File · Nature Communications]

Reviewers' comments:

Reviewer #1 (Remarks to the Author):

In the manuscript titled "On-chip AP-MS: Interaction proteomics on a microfluidics platform", Furlan et al., present a microfluidistic platform for the AP-MS, which in principle allows performing analyses from a miniscule number (10-100k) of cells. I really like this the approach and see great potential in it. However, in the current form and with the provided examples, it is extremely difficult to judge if this is just another proof-of-concept or actual and viable product/method. Several important issues should be addressed prior to publishing.

Major quibbles:

The authors chose to used GFP-fusions and GFP nanobody-conjugated magnetic beads in their approach. Why this approach was chosen over Strep-tag and streptavidin, which is the current "gold standard" in AP-MS? The authors state that their "platform can be adapted to purify protein complexes using antibodies in conjunction with protein A/G beads", why not with Strep-tag and streptavidin? This option or caveat would need to be discussed in detail in the manuscript. I would love to see this approach adapted to Strep-tagged proteins (or BioID), and an experiment showing this would strengthen the impact of the manuscript significantly.

The authors candidly used SMC1A, subunit of the cohesin complex, as bait. This is well studied and extremely stable small complex and therefore used in many studies for method validation. Due to this the results for cohesion complex actually look "too good", even on quantitative perspective. With CCC complex the results also look good on the subunit identifications, however the assessment of quantitative reproducibility or comparison with different cell amounts is made difficult due to the lack of actual LFQ values and visualization of the LFQ values as log₂FE values. The LFQ values should be shown as a supplementary table. Additionally, the authors should compare their CCC complex results to that of the Hein et al., 2015, Cell paper.

To really benchmark their approach the authors should use Mediator complex for testing, similarly as several articles before (PMID: 20647355, PMID: 17138671, PMID: 28240253, PMID: 23602568, PMID: 29568061, PMID: 26496610). Any of the MED12, MED13, MED11 or CDK8 would be a good candidate

Minor quibbles

When using AP-MS, regulator subunits PDS5 and WAPL are often detected with cohesin complex (for example Hein et al., 2015, Cell). Were these interactions missed even with when 500ug input material was used? This leads some doubts of the actual sensitivity of the method.

The MS parameters are extensively described; however, the MS database search parameters are almost completely missing. These should be added.

Reviewer #2 (Remarks to the Author):

The authors describe a low-volume microfluidic sample processing platform based on multilayer soft lithography for processing small samples for LC-MS based protein-protein interaction studies. The workflow enables a 50–100-fold reduction in sample input to just 4 μg or $\sim 12,000$ cells, which enables studies in sample-limited applications that were not previously possible. The work is significant and the results are convincing. The manuscript is presented in a letter format, which may not be the most appropriate for Nature Communications. The authors may consider “unpacking” some of the supplementary data and methods into the main text. Aside from that minor item, I recommend accepting this article as is.

We were very happy with the constructive and positive comments the reviewers made on our manuscript "**On-chip AP-MS: Interaction proteomics on a microfluidics platform**". We have revised the manuscript according to the suggestions brought forward by you and the reviewers. Please find below a detailed response to all comments.

Reviewer #1

The authors chose to use GFP-fusions and GFP nanobody-conjugated magnetic beads in their approach. Why this approach was chosen over Strep-tag and streptavidin, which is the current "gold standard" in AP-MS? The authors state that their "platform can be adapted to purify protein complexes using antibodies in conjunction with protein A/G beads", why not with Strep-tag and streptavidin? This option or caveat would need to be discussed in detail in the manuscript. I would love to see this approach adapted to Strep-tagged proteins (or Biold), and an experiment showing this would strengthen the impact of the manuscript significantly.

We thank the reviewer for making this valid point. GFP is a valuable and versatile tag that has been broadly used to study proteins for more than 2 decades, both by microscopy and biochemical techniques. Many laboratories tag their proteins of interest with GFP, and as a matter of fact and *Hein et al (Cell, 2015)* used the same tagging approach as we do in our manuscript (BAC-GFP transgenes) to characterize the human interactome. In the Vermeulen lab, our mainstream affinity purification workflows rely on the use of the GFP tag given the fact that high affinity nanobodies for GFP have been developed, both commercially (www.chromotek.com) and non-commercially (<https://www.ncbi.nlm.nih.gov/pubmed/26436480>). These nanobodies are highly suitable in combination with on-bead digestion workflows given the small size (12 kDa) of the nanobodies compared to for example Flag-M2 beads which are coated with native heavy and light chains. We agree with Reviewer #1 that our platform could in principle also be used for baits containing other tags (i.e. HA, Strep-tag, biotin). However, the design of the chips that we use for our experiments requires using beads with a size between 2 and 20 μm , while Streptactin and Flag/HA beads are not available in this size.

Nevertheless, to follow up on this suggestions of the reviewer and to further illustrate the broad applicability of our On-chip AP-MS platform, we now included in the revised manuscript the purification of the human cohesin complex from 4 μg of HeLa Kyoto extract using a commercially available SMC3 antibody immobilized to protein A/G beads (Fig. 4 of the revised manuscript). The application of an antibody against a native protein represents a critical further step towards the use of clinical samples or primary cells on our On-chip AP-MS platform, as these samples are often not compatible with transgenic tagging technologies. Notably, these experiments will always depend on the quality of the antibodies as well as the expression level of the targeted bait. In any case, with the development of On-chip AP-MS, the field of mass spectrometry-based interaction proteomics is entering a new era of applications in biology and medicine, which thus far, due to technical constraints, have remained unexplored.

The authors candidly used SMC1A, subunit of the cohesin complex, as bait. This is well studied and extremely stable small complex and therefore used in many studies for method validation. Due to this the results for cohesion complex actually look "too good", even on quantitative perspective. With CCC complex the results also look good on the subunit identifications, however the assessment of quantitative reproducibility or comparison with different cell amounts is made difficult due to the lack of actual LFQ values and

visualization of the LFQ values as log₂FE values. The LFQ values should be shown as a supplementary table. Additionally, the authors should compare their CCC complex results to that of the Hein et al., 2015, Cell paper.

We thank the reviewer for his/her kind words concerning our Cohesin complex purifications. We appreciate this concern raised by the Reviewer. In the revised manuscript, we included (i) a supplementary table containing the original data for each experiment (MaxQuant Protein groups file); (ii) the filtered data (processed in R and Perseus); and (iii) the iBAQ values used for stoichiometry estimation. We note that the stoichiometry values we obtained for the CCC complex are in good agreement with the stoichiometry values published by Hein et al., (Cell 2015), as shown in Figure 2h in the revised manuscript.

To really benchmark their approach the authors should use Mediator complex for testing, similarly as several articles before (PMID: 20647355, PMID: 17138671, PMID: 28240253, PMID: 23602568, PMID: 29568061, PMID: 26496610). Any of the MED12, MED13, MED11 or CDK8 would be a good candidate.

We appreciate this comment and performed the experiments as suggested by the reviewer to further benchmark our method. We purified the Mediator complex from 25 x 10³ HeLa Kyoto cells expressing a CDK8-GFP BAC transgene as bait. This experiment yielded essentially all Mediator subunits, with results comparable to those obtained in the publications mentioned above by the reviewer which were performed with quite large amounts of input material (Fig. 3 of the revised paper). These results further underline the excellent performance of our On-chip AP-MS method.

Minor quibbles

When using AP-MS, regulator subunits PDS5 and WAPL are often detected with cohesin complex (for example Hein et al., 2015, Cell). Were these interactions missed even with when 500 µg input material was used? This leads some doubts of the actual sensitivity of the method.

We thank the reviewer for this comment. We do detect interactions between the core cohesin complex and PDSA/B and WAPAL when using 500 µg of starting material, albeit at very low stoichiometry (<0.1, see Supplementary Figure 1a). These interactions, however, are lost when the starting material is reduced to 100 µg of input extract (see Supplementary Figure 1b). Based on these results we conclude that our low input AP-MS method has its limitations regarding the ability to detect substoichiometric (< 0.1 relative to the bait) and/or dynamic interactions between core subunits and regulatory proteins for any given protein complex, obviously depending on the stoichiometry, affinity, and copy number of the bait and its partners.

The MS parameters are extensively described; however, the MS database search parameters are almost completely missing. These should be added.

Raw mass spec data was searched using the MaxQuant software with default settings. As requested by Reviewer #1, we now extended our methods section to include database search parameters further facilitating to replicate our data analysis.

Reviewer #2

The authors describe a low-volume microfluidic sample processing platform based on multilayer soft lithography for processing small samples for LC-MS based protein-protein interaction studies. The workflow enables a 50–100-fold reduction in sample input to just 4 µg or ~12,000 cells, which enables

studies in sample-limited applications that were not previously possible. The work is significant and the results are convincing. The manuscript is presented in a letter format, which may not be the most appropriate for Nature Communications. The authors may consider “unpacking” some of the supplementary data and methods into the main text. Aside from that minor item, I recommend accepting this article as is.

We thank Reviewer #2 his/her kind words. We believe that a relatively short format suits this manuscript very well and we are somewhat reluctant to incorporate numerous supplementary figures into the main paper. However, we are open to further suggestions by the editor to present our data in the best possible way.